# Early Administration of Tolvaptan Can Improve Survival in Patients with Cirrhotic Ascites

**DOI:** 10.3390/jcm10020294

**Published:** 2021-01-14

**Authors:** Atsushi Hosui, Takafumi Tanimoto, Toru Okahara, Munehiro Ashida, Kohsaku Ohnishi, Yuhhei Wakahara, Yukihiro Kusumoto, Toshio Yamaguchi, Yuka Sueyoshi, Motohiro Hirao, Takuya Yamada, Naoki Hiramatsu

**Affiliations:** Department of Gastroenterology and Hepatology, Osaka-Rosai Hospital, 1179-3 Nagasone, Kitaku, Sakai, Osaka 591-8025, Japan; aijyousaizu@yahoo.co.jp (T.T.); t.okahara@osakah.johas.go.jp (T.O.); mashida1013@yahoo.co.jp (M.A.); k-ohnishi@osakah.johas.go.jp (K.O.); youu0805hey@yahoo.co.jp (Y.W.); yukizarulucky_3lib@yahoo.co.jp (Y.K.); yamaguchi-to@osakah.johas.go.jp (T.Y.); sueyoshi@osakah.johas.go.jp (Y.S.); snowmotty@yahoo.co.jp (M.H.); yamada@osakah.johas.go.jp (T.Y.); hiramatsu@osakah.johas.go.jp (N.H.)

**Keywords:** tolvaptan, cirrhotic ascites, survival rate, furosemide

## Abstract

(1) Backgrounds and aim: Tolvaptan, a selective vasopressin type 2 receptor antagonist, was approved for ascites, and its short-term efficacy and safety have been confirmed. However, it is still unclear whether this novel drug may improve long-term survival rates in cirrhotic patients with ascites. (2) Patients and methods: A total of 206 patients who responded insufficiently to conventional diuretics and were hospitalized for refractory ascites for the first time were retrospectively enrolled in this study. Among them, the first 57 consecutive patients were treated with conventional diuretics (the conventional therapy group); the latter 149 consecutive patients were treated with tolvaptan in addition to the conventional therapy (the tolvaptan group). (3) Results: The exacerbation of renal function was significantly milder in the tolvaptan group than in the conventional therapy group. The prognostic factors for survival in the tolvaptan group were being male, having hyperbilirubinemia, having a high blood urea nitrogen (BUN), and receiving high-dose furosemide at the start of tolvaptan treatment. The one-year and three-year cumulative survival rates were 67.8 and 45.3%, respectively, in patients with low-dose furosemide (<40 mg/day) at the start of tolvaptan treatment. The prognosis was significantly better in the tolvaptan group with low-dose furosemide than in the conventional therapy group (*p* < 0.001). (4) Conclusion: Tolvaptan can improve survival in patients with cirrhotic ascites, especially when tolvaptan is started before high-dose furosemide administration.

## 1. Introduction

Progression of liver diseases is characterized by a large decrease in the excretion of urinary sodium and accumulation of retained fluid within the abdominal cavity. For patients with liver cirrhosis who have ascites, current guidelines recommend the administration of a diuretic drug if the efficacy of sodium intake restriction is inadequate [1,2]. Conventional diuretics are natriuretic drugs that block sodium reabsorption in the nephrons, increasing renal sodium excretion to achieve a negative sodium balance [3,4]. Although ascites can be controlled through the restriction of sodium intake and administration of a natriuretic medication, some patients with ascites develop resistance to conventional therapy, which is referred to as refractory ascites (RA). For the treatment of diuretic-intractable ascites, an effective diuretic dosage has not yet been determined because of the development of severe diuretic-related side effects [5]. The strategy for treating ascites refractory to diuretic therapy has still not been established.

Recently, several studies have evaluated the effects of aquaretic drugs, such as tolvaptan, for treating ascites resistant to conventional diuretics [6,7]. Tolvaptan, which blocks arginine vasopressin (AVP) from binding to V_2_ receptors in the distal nephrons and thus restricts water reabsorption, is an ideal aquaretic drug for the treatment of hyponatremia in conditions associated with increased circulating levels of antidiuretic hormones, such as decompensated liver cirrhosis [8,9]. Tolvaptan was approved on September 2013 in Japan, and many investigators have reported that tolvaptan is effective against RA for a short time and have shown many parameters predictive of its effectiveness [10,11,12]. However, it remains unclear whether a combination therapy with natriuretic and aquaretic medications is more effective than the conventional therapy with a natriuretic medication for patients with liver cirrhosis who have ascites for an extended time. It is also unknown when tolvaptan should be used in combination with conventional diuretics. Administration of conventional diuretics for an extended time causes activation of the renin-aldosterone system and, finally, worsens renal function. The one-year probability of survival after developing RA was reported to be approximately 30%, and the mean survival was only seven ± two days in patients developing hepatorenal syndrome [13]. It is important that patients with RA do not develop renal dysfunction. To clarify these issues, we compared the effects of combination diuretic therapy with conventional diuretic therapy in cirrhotic ascites patients.

## 2. Materials and Methods

The primary outcome was the overall survival, and the secondary outcomes were the prognostic factors for survival and contributing factors to a good response to tolvaptan. A total of 206 patients who responded insufficiently to conventional diuretics and who were hospitalized for RA for the first time were retrospectively enrolled in this study. The first 57 consecutive patients were treated with conventional diuretics and intravenous albumin administration between January 2010 and November 2013 (for approximately four years) in the conventional therapy group; a historical control was used for the group without administration of tolvaptan, which we treated only with conventional diuretics. The latter 149 consecutive patients were treated with tolvaptan in addition to the conventional therapy between December 2013 and December 2018 (for five years) in the tolvaptan group. In the conventional therapy group, the dose of furosemide or spironolactone was basically increased; sometimes, ascitic fluid was removed and a human serum albumin preparation was dripped intravenously. In the tolvaptan group, tolvaptan was administered without increasing conventional diuretics, and all other treatments were identical to those in the conventional therapy group. The initial administration dose of tolvaptan was 3.75 mg, and the dose was increased to 7.5 mg if ascites was not improved. We usually continue to treat tolvaptan, thus, the duration of treatment is the same as the observation period. The initial therapeutic effect of tolvaptan is defined as the “body weight [decreasing] by 1.5 kg or more within a week from the start of tolvaptan administration” [14]. The tolvaptan group was next divided into two groups according to the administration dose of furosemide on admission (<40 mg/day, low-dose furosemide group; ≥40 mg/day, high-dose furosemide group) (Figure 1).

The study observation period was from 1 January 2010 to 31 December 2019. The starting point was the hospitalization day. The conventional therapy group was treated with increasing conventional diuretics and/or administration of intravenous albumin and/or removing ascites (*n* = 57). The tolvaptan group was started on tolvaptan without increasing conventional therapy (*n* = 149).

Liver function was examined at least every two months, and imaging (computed tomography, ultrasound, or magnetic resonance imaging) results were evaluated every three months. This study was approved by the ethics committee of Osaka-Rosai Hospital.

Data were analyzed using the statistical software JMP 11.0.1 (SAS Institute, Tokyo, Japan), and the data are presented as means ± SEs. Data from the two groups were compared using unpaired *t*-tests. Multiple comparisons were performed by the Cox proportional hazards regression test; *p* < 0.05 was considered statistically significant. The log-rank test was used to assess the cumulative incidence rates for survival.

## 3. Results

### 3.1. Changes in Renal Function after Hospitalization for the First Time in the Conventional Therapy Group and the Tolvaptan Group

The characteristics of the 206 patients with ascites treated with diuretics are shown in Table 1. A higher BUN value and lower sodium concentration were observed in the tolvaptan group than in the conventional group, and no significant differences aside from these factors were found in the clinical backgrounds of patients in the two groups. For safety, adverse events (AEs) were found to be similar in approximately 20% of the patients in each group, but different types of AEs were observed; the conventional group had renal dysfunction (12.2%), hepatic encephalopathy (5.2%), and general fatigue (3.5%), while the tolvaptan group had thirst (8.7%), general fatigue (5.3%), and appetite loss (4.0%). We next evaluated changes in renal function after hospital admission due to RA, because patients with ascites gradually become unresponsive to conventional diuretics, followed by renal dysfunction after administration of furosemide and spironolactone. Both the BUN and creatinine values gradually increased in the conventional therapy group over the course of one year, but they remained at an almost normal range in the tolvaptan group (Figure 2). The occurrence of hepatorenal syndrome was 58% and 11% during one year, in the control group and the tolvaptan group, respectively (*p* < 0.001). Tolvaptan was reported to cause hypernatremia after treatment, and thus, changes in sodium were examined, but sodium levels did not change at all in these two groups.

### 3.2. Overall Survival in the Conventional Therapy Group and the Tolvaptan Group

We investigated the overall survival (OS) regarding the admission day as the starting point in the two groups and found that OS at one and two years in all patients were 43.6% and 30.6%, respectively. As shown in Figure 3, OS at one and two years in the tolvaptan group and in the conventional therapy group were 46.2% and 35.4%, and 36.8% and 19.9%, respectively. The prognosis was statistically insignificant between these two groups (*p* = 0.38).

### 3.3. A Critical Contributory Factor to the Good Response to Tolvaptan

Not all patients with ascites respond to tolvaptan, and many predictive factors have been reported. A good response to tolvaptan is defined as a 1.5 kg decrease in body weight after one week of treatment, as shown above. The good response rate based on this definition was 65.8%. We investigated predictive factors and univariately analyzed age, gender, and laboratory data (including AVP, aldosterone, renin, urine osmolality (U osm), and plasma osmolality (P osm), and found that five factors (creatinine, BUN, potassium, Child-Pugh score, and administration period of conventional diuretics) were positively associated (Table 2). A multivariate analysis revealed that only low BUN (<20 mg/dL: *p* = 0.005) was a critical contributory factor to the good response to tolvaptan, which was consistent with the findings of the START study [11].

### 3.4. The Prognostic Factors for Survival in the Tolvaptan Group

We next examined the prognostic factors among age, gender, renal and liver functions, administration dose of furosemide or spironolactone, and the presence/absence of HCC in the tolvaptan group. As shown in Table 3, a univariate analysis revealed that the prognostic factors were being male, having hyperbilirubinemia, having a high BUN, receiving high-dose furosemide at the start of tolvaptan treatment, and the presence of hepatocellular carcinoma (HCC). Finally, a multivariate analysis of these five factors clarified that the prognostic factors were being male (OR 1.59, *p* = 0.049), having hyperbilirubinemia (>2 mg/dL, OR 1.89, *p* = 0.009), having a high BUN (>25 mg/dL, OR 1.67, *p* = 0.031), and receiving high-dose furosemide (≥40 mg/day, OR 2.63, *p* < 0.001).

### 3.5. The Prognosis in the Tolvaptan Group with Low- or High-Dose Furosemide and the Conventional Therapy Group

The dose of furosemide administered was one of the most important prognostic factors, thus, the tolvaptan group was divided into two groups according to the dose of furosemide at the start of tolvaptan treatment (<40 mg/day, low-dose furosemide group; ≥40 mg/day, high-dose furosemide group). The clinical backgrounds of these two groups and the conventional therapy group are shown in Table 4. Renal function had already worsened in the high-dose furosemide group compared to that in the other groups, and there were no differences except renal function between the low-dose furosemide group and the conventional therapy group.

As shown in Figure 4, OS at one and two years in the tolvaptan group with low-dose furosemide and in the conventional therapy group were 67.8% and 52.8% and 36.8% and 19.9%, respectively. The prognosis was significantly better in the low-dose furosemide group than in the conventional therapy group (log-rank test *p* < 0.0001).

This result shows the possibility that tolvaptan can improve survival in patients with cirrhotic ascites, especially in those whose tolvaptan treatment was started before high-dose furosemide administration.

The overall survival rate was better in the tolvaptan group with low-dose furosemide than in the conventional therapy group or in the tolvaptan group with high-dose furosemide (log-rank test *p* < 0.001).

## 4. Discussion

We have experienced difficulties with and inadequacies in conventional diuretic treatments in patients with cirrhotic ascites. Although the deterioration of renal function is frequently observed with conventional diuretic therapy, renal dysfunction has been shown to be related to poor prognosis. The therapeutic strategy for cirrhotic ascites has changed drastically since tolvaptan was approved in 2013 [15]. Tolvaptan, an oral AVP V2 receptor antagonist, has been used as a new diuretic for ascites in combination with conventional diuretics. It is unknown whether this novel drug may help with the maintenance of renal function and improve long-term survival rates in cirrhotic patients with ascites. Tolvaptan has been reported to be quite effective against RA for a short time [7,15]. Thus, it was difficult to do a randomized controlled trial for a longer time.

The prognosis was statistically insignificant between the conventional therapy group and the tolvaptan group in this study. There were two reasons for the negative result. First, the tolvaptan group had a higher BUN value and lower serum sodium than those in the conventional therapy group. A higher BUN was associated with renal dysfunction. Low serum sodium was also related to an increased risk of death in patients with cirrhosis [16]. These clinical backgrounds might adversely affect the result. Second, a higher BUN showed not only renal dysfunction, but also unresponsiveness to tolvaptan. Tolvaptan is quite effective, but not all patients with cirrhotic ascites respond to this new drug. As BUN values worsened, patients were less likely to respond to tolvaptan [11]. The patients with higher BUN levels are presumed not to have enough water in their blood vessels to be excreted by tolvaptan, or to have worse renal perfusion, thus, tolvaptan is not effective in patients with higher BUN levels [17].

OS among patients with HCC and without HCC in this cohort was investigated, and we found that it is significantly better in the HCC (−) group than in the HCC (+) group (Appendix A). However, the prognostic factor among patients with refractory ascites was gender (male), liver function (high value of total bilirubin), renal function (high value of BUN), and administration dose of furosemide (high dose of furosemide), but not the presence of HCC in the multivariate analysis. Patients with refractory ascites have poor liver and renal function, and, in that case, the prognosis might not depend on the presence/absence of HCC.

Tolvaptan can control RA, and, as a result, it leads to a good prognosis when it is started before high-dose furosemide administration. There are some reasons why tolvaptan improves OS in patients with ascites. First, it can decrease hepatic ascites and body fluid retention without worsening renal function, as shown in this manuscript. Before tolvaptan can be used, we had to increase the dose of conventional diuretics, which was finally followed by renal dysfunction. Second, tolvaptan can control RA more quickly than the conventional therapy, and, as a result, the nutritional status of patients may rapidly improve. Skeletal muscle mass decreases with age, especially in patients with liver cirrhosis (LC). Sarcopenia is characterized by the loss of muscle mass and is significantly associated with mortality in patients with LC [18]. Improvement in nutritional status can suppress the reduction of muscle mass, and it might be related to a good prognosis. On the other hand, loop diuretics directly suppress the differentiation of myofibroblasts, and sarcopenia in patients with LC may be attributable to treatment with loop diuretics [19]. Third, as tolvaptan can control ascites, the number of complications of decompensated cirrhosis (such as spontaneous bacterial peritonitis) decreases, and patients do not need to be in the hospital.

Not all patients with ascites respond to tolvaptan. The effectiveness rate has been reported to be approximately 50–60%, and non-responders have a poor prognosis, which is similar to those who are treated with conventional diuretics [11]. The present study showed that tolvaptan should be started before the BUN value becomes too high in order to reduce the number of non-responders. This high BUN value implies not only renal dysfunction, but also the deterioration of osmotic pressure in the renal interstitium, which is mainly caused by the administration of high-dose loop diuretics. Thus, we should be careful not to give high-dose loop diuretics in order to obtain a good response to tolvaptan.

Some investigators have already reported the efficacy and safety of tolvaptan. Uchida has shown that the ascites-related events-free duration was prolonged following tolvaptan treatment compared with that before treatment [20]. Kogiso also explained the impact of continued administration of tolvaptan and proved that long-term tolvaptan treatment increased serum levels of albumin, decreased ammonia levels, and preserved renal function [21]. We have never found a study in which the OS was compared between the tolvaptan group and the conventional therapy group in more than 200 patients with refractory ascites for more than four years.

## 5. Conclusions

This is the first report to reveal that tolvaptan can improve survival, especially when tolvaptan treatment is started before high-dose furosemide administration. Therapy for RA has drastically changed, and clinicians give tolvaptan early for RA rather than increasing the doses of conventional diuretics. The limitations of this study are the small number of patients and the retrospective nature of this study. Prospective and double-blind studies should be better, but tolvaptan has been reported to be quite effective in short-term and long-term administration. It might be difficult to prove it by prospective study.

## Figures and Tables

**Figure 1 jcm-10-00294-f001:**
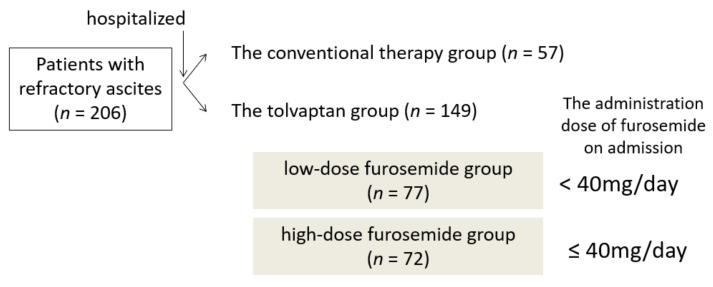
Scheme of this study.

**Figure 2 jcm-10-00294-f002:**
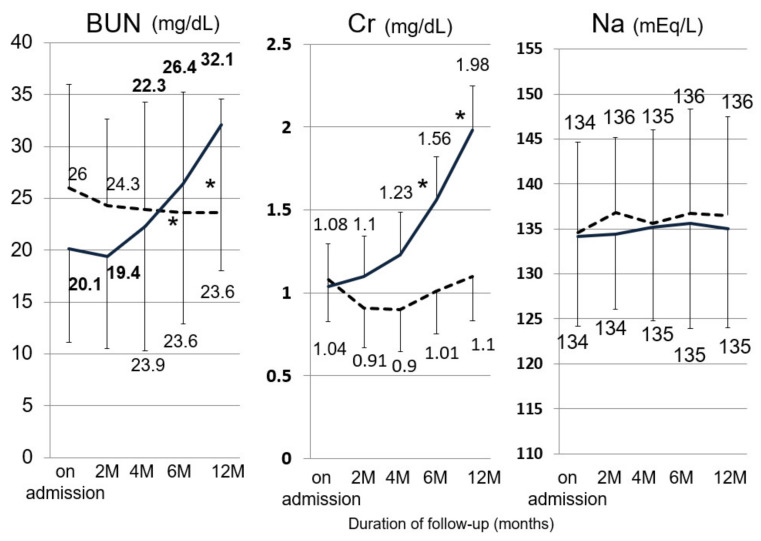
Serial changes in the serum concentration of BUN, creatinine, and sodium in the conventional therapy group and the tolvaptan group. Solid line: the conventional therapy group; dotted line: the tolvaptan group. Asterisks indicate significant differences (* *p* < 0.01 compared with the value recorded before treatment).

**Figure 3 jcm-10-00294-f003:**
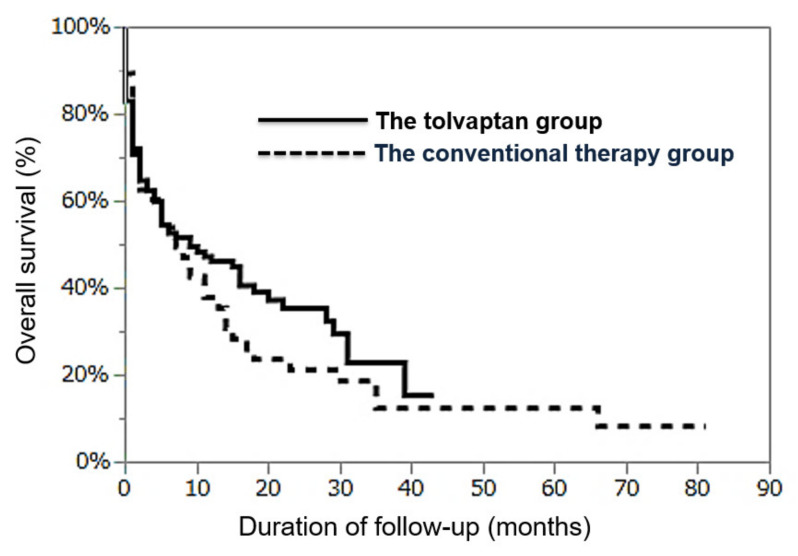
Comparison of cumulative survival rates between the conventional therapy group and the tolvaptan group.

**Figure 4 jcm-10-00294-f004:**
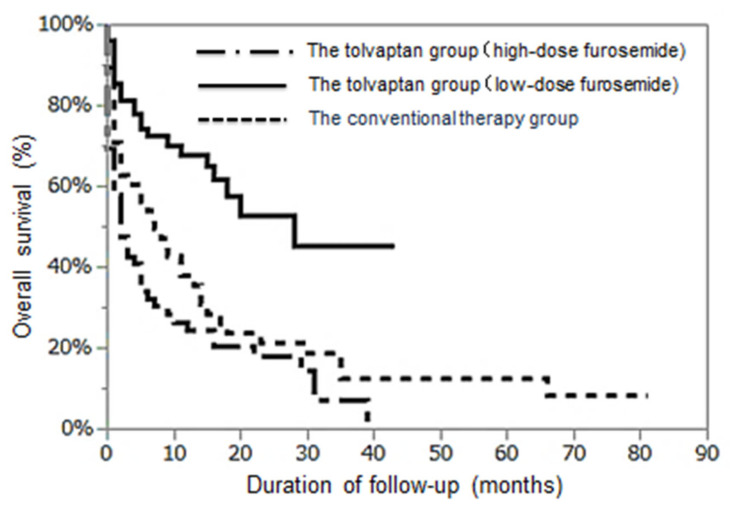
Comparison of cumulative survival rates among the conventional therapy group and the tolvaptan groups with low- or high-dose furosemide.

**Table 1 jcm-10-00294-t001:** Clinical backgrounds in the conventional therapy group and the tolvaptan group enrolled in this study.

	Conventional Group	Tolvaptan Group	*p*-Value
Cases	57	149	
Mean Age	70.7 ± 11.3	72.6 ± 10.4	0.26
Gender (Male/Female)	33/24	94/55	0.47
AST (U/L)	67.1 ± 33.2	66.3 ± 20.0	0.97
ALT (U/L)	44.2 ± 30.3	44.5 ± 50.7	0.98
gGTP (U/L)	87.1 ± 104.5	61.9 ± 80.0	0.44
ALP (IU/mL)	418 ± 183	486 ± 280	0.43
Total Bilirubin (mg/dL)	1.92 ± 1.40	2.67 ± 4.51	0.23
Albumin (g/dL)	2.79 ± 0.42	2.69 ± 0.45	0.15
PT Activity (%)	57.8 ± 16.7	59.9 ± 17.4	0.45
Creatinine (mg/dL)	1.04 ± 0.91	1.08 ± 0.53	0.70
BUN (mg/dL)	20.1 ± 11.9	26.0 ± 18.9	**0.028**
Na (mEq/L)	137.1 ± 2.4	134.3 ± 3.9	**0.021**
K (mEq/L)	3.96 ± 0.72	4.09 ± 0.82	0.63
AFP (ng/mL) Median	232	159	0.78
PIVKA-2 (mAU/mL) Median	227	274	0.40
Child-Pugh Score	8.8 ± 1.3	8.7 ± 1.3	0.75
Child-Pugh Status (B/C)	31/26	83/66	0.75
Administration Dose of Furosemide (mg)	34.2 ± 35.6 (0–240)	29.3 ± 22.3 (0–120)	0.23
Administration Dose of Spironolactone (mg)	34.7 ± 25.8 (0–100)	28.8 ± 22.2 (0–100)	0.11
Administration Period of Conventional Diuretics (Months)	24.2 ± 21.3	22.8 ± 29.2	0.43
HCC (Past History)	19	41	0.13
HCC (First Six Months)	3	8	0.95
HCC (Presence/Absence)	33/24	89/60	0.06

Marked with bold the *p*-value of BUN and Na are significantly different among two groups. Abbreviations: AST, aspartate aminotransferase; ALT, alanine amino transferase; gGTP, γ-glutamyl transpeptidase; ALP, alkaline phosphatase; PT, prothrombin; BUN, blood urea nitrogen; AFP, α-fetoprotein; PIVKA-2, protein induced by vitamin K absence; HCC, hepatocellular carcinoma.

**Table 2 jcm-10-00294-t002:** Contributing factors to a good response to tolvaptan. A good response to tolvaptan is defined as a 1.5 kg decrease in body weight after one week of treatment.

	Univariate Analysis		Multivariate Analysis	
	OR	*p*-Value	OR	*p*-Value
Age (Older Than 70 Years)	2.02	0.07		
Gender (Female)	1.16	0.675		
Total Bilirubin (<2 mg/dL)	1.97	0.234		
Albumin (>2.8 g/dL)	1.82	0.10		
PT Activity (>70%)	1.23	0.56		
Creatinine (<1.1 mg/dL)	3.12	0.0015	1.42	0.46
BUN (<25 mg/dL)	4.43	<0.0001	3.59	0.005
Na (>135 mEq/L)	2.10	0.1050		
K (<4.0 mEq/L)	2.58	0.0416	1.12	0.65
U osm (≤400 mOSM/L)	1.71	0.3049		
P osm (>280 mOSM/L)	1.04	0.9398		
AVP (≤2.5 pg/mL)	1.83	0.3967		
Aldosterone (≤200 pg/mL)	1.40	0.6128		
Renin Activity (≤5.0 ng/mL/h)	1.08	0.9021		
Child-Pugh Score (≤10)	2.34	0.0297	1.94	0.20
Administration Dose of Furosemide (≤40 mg)	1.48	0.267		
Administration Dose of Spironolactone (≤50 mg)	1.02	0.967		
Administration Period of Conventional Diuretics (<Two Years)	2.76	0.0462	1.72	0.24
HCC (Presence)	2.03	0.058		

Abbreviations. PT, prothrombin; BUN, blood urea nitrogen; U osm, urine osmolality; P osm, plasma osmolality; AVP, arginine vasopressin; HCC, hepatocellular carcinoma.

**Table 3 jcm-10-00294-t003:** The prognostic factors for survival in the tolvaptan group.

	Univariate Analysis		Multivariate Analysis	
	OR	*p*-Value	OR	*p*-Value
Age (Older Than 70 Years)	1.40	0.16		
Gender (Male)	1.42	0.013	1.59	0.049
Total Bilirubin (>2 mg/dL)	1.73	0.0158	1.89	0.009
Albumin (<2.8 g/dL)	1.39	0.15		
PT Activity (<70%)	1.56	0.076		
Creatinine (>1.1 mg/dL)	1.43	0.12		
BUN (>25 mg/dL)	1.67	0.024	1.67	0.031
Administration Dose of Furosemide (≥40 mg)	3.20	< 0.001	2.63	< 0.001
Administration Dose of Spironolactone (≥50 mg)	1.32	0.22		
HCC (Presence)	1.73	0.0164	1.47	0.11

Abbreviations: PT, prothrombin; BUN, blood urea nitrogen; HCC, hepatocellular carcinoma.

**Table 4 jcm-10-00294-t004:** Clinical backgrounds in the two tolvaptan groups. The tolvaptan group was divided into two groups according to the dose of furosemide administered on admission (<40 mg/day, low-dose furosemide group; ≥40 mg/day, high-dose furosemide group).

	Tolvaptan Group		*p*-Value
	High-Dose Furosemide	Low-Dose Furosemide	
Cases	72	77	
Mean Age	72.8 ± 10.3	72.3 ± 10.5	ns
Gender (Male/Female)	44/28	50/27	ns
Total Bilirubin (mg/dL)	2.69 ± 3.43	2.66 ± 5.34	ns
Albumin (g/dL)	2.64 ± 0.49	2.73 ± 0.39	ns
PT Activity (%)	59.3 ± 17.1	60.4 ± 17.7	ns
Creatinine (mg/dL)	1.14 ± 0.54	1.02 ± 0.52	ns
BUN (mg/dL)	29.9 ± 21.1	22.4 ± 15.8	0.014
Administration Dose of Furosemide (mg)	45.8 ± 15.2	16.5 ± 7.4	<0.001
Administration Dose of Spironolactone (mg)	33.3 ± 26.2	24.5 ± 16.5	0.0015
HCC (Presence/Absence)	49/23	40/37	ns

Abbreviations: PT, prothrombin; BUN, blood urea nitrogen; HCC, hepatocellular carcinoma.

## Data Availability

Data is contained within the article or Appendix A.

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
