# Peer review of "Early Administration of Tolvaptan Can Improve Survival in Patients with Cirrhotic Ascites"

_jcm, 2021, doi:10.3390/jcm10020294_

Round 1

Reviewer 1 Report

The paper attempted  to prove  the beneficial effect of the use of Tolvaptan (selective vasopressin type 2 receptor antagonist) in addition to the  diuretic treatment,  on the long-term survival rates in cirrhotic patients with ascites.  The authors showed that the 1-year/3-year cumulative survival rates were 67.8/45.3% in patients with low-dose furosemide (<40 mg/day) at the start of Tolvaptan treatment and 24.4/3.1% in patients with high-dose furosemide at the start of Tolvaptan treatment (≥40 mg/day). The Authors concluded that Tolvaptan can improve survival in patients with cirrhotic ascites, especially in those whose Tolvaptan treatment started before high-dose furosemide administration. The topic is certainly important, however, in the reviewer’s opinion the paper as it now stands is not ready to be published, as there are important aspects to be re-analysed in the work.

Major revisions

  • Patients and methods: In this study were enrolled retrospectively 206 patients with refractory ascites from January 2010 to December 2018.  Patients enrolled in the first 4 years underwent conventional diuretic treatment and patients enrolled from December 2013 to December 2018 used Tolvaptan added to conventional diuretic treatment.  It is not clear if the patients included retrospectively in 2013 and 2014  were also enrolled in clinical trials.  In this regard an essential information missing in the paper are: i) the date of approval of the use  of Tolvaptan  for treatment of refractory ascites in Japan,   ii) the Tolvaptan dosage, iii)  the duration of the treatment.  Those data are essential to understand in full the methodology adopted by the authors and to support their conclusion. Those aspects need to  addressed in the revised version of the paper.
  • Results:
    • Table 1:  the Authors compared  the “Conventional Group” encompassing  57 patients with the “Tolvaptan Group” encompassing 149 patients.  The population of those two groups  is unbalanced, it  would be better in the reviewer opinion to split the “Tolvaptan Group” in two sub-groups according to their diuretic treatment as done by the authors in following analyses.
    • Table 1: the AFP median value in  the “Conventional Group” was 232 ng/ml while in the “Tolvaptan Group” it was 159 ng/ml. HCC was present in 33 patients of   the “Conventional Group” while it was present  in 89 patients of the “Tolvaptan Group”.  How many patients had a previous diagnosis of HCC and how many patients developed HCC during the first six months of the follow-up period in the three different groups (i.e. Conventional Treatment, High-dose furosemide and Low-dose  of furosemide treatment)? 
    • At the time of enrolment, how many patients were in Child-Pug class B and how many in class C?
    • As reported by the Authors, the 1-year probability of survival after developing RA was to be approximately 30%. Would be, therefore, useful  to highlight how many patients underwent to Transjugular intrahepatic portosystemic shunts and/or to liver transplantation during the follow-up period. Comments in this regard are meritorious and recommended in the revised version of the paper.
    • In the comparison of the cumulative survival rates between the “Conventional therapy Group” and 126 the "Tolvaptan Group" section,   Figure 3 showed  a survival period of 90 months.  The period of enrolment, however, was from 2010 to 2018. In  the reviewer opinion  would be better  to analyse and represent  the survival rates at 12 and 24 months for all patients. 

Author Response

Thank you for good suggestion and advice.

Major revisions

  • Patients and methods: In this study were enrolled retrospectively 206 patients with refractory ascites from January 2010 to December 2018.  Patients enrolled in the first 4 years underwent conventional diuretic treatment and patients enrolled from December 2013 to December 2018 used Tolvaptan added to conventional diuretic treatment.  It is not clear if the patients included retrospectively in 2013 and 2014 were also enrolled in clinical trials. 

→Thank you for good suggestion and advice. The 57 consecutive patients were treated with conventional diuretics between January 2010 and November 2013 (approximately 4 years), and 149 patients were treated with tolvaptan between December 2013 and December 2018 (about 5 years).  We corrected the text not to misunderstand the manuscript in the Patients and methods section (page 2, lines 68 and 72).

In this regard an essential information missing in the paper are: i) the date of approval of the use of Tolvaptan for treatment of refractory ascites in Japan, ii) the Tolvaptan dosage, iii) the duration of the treatment.  Those data are essential to understand in full the methodology adopted by the authors and to support their conclusion. Those aspects need to addressed in the revised version of the paper.

→As the reviewer suggested, these are essential information. Thank you.

Tolvaptan was approved on September 2013, and we started to treat patients with tolvaptan (page 2, lines 48-49).  Initial administration dose of tolvaptan was 3.75mg, and the dose was increased to 7.5mg if ascites was not improved.  We usually continue to treat tolvaptan, thus the duration of treatment is the same as observation period (page 2-3, lines 77-80).  We added these sentences in the Patients and methods section.

  • Results:
    • Table 1:  the Authors compared the “Conventional Group” encompassing 57 patients with the “Tolvaptan Group” encompassing 149 patients.  The population of those two groups is unbalanced, it would be better in the reviewer opinion to split the “Tolvaptan Group” in two sub-groups according to their diuretic treatment as done by the authors in following analyses.

→Yes, that’s true. But there were no differences between these two groups except the value of BUN and serum sodium.  A higher BUN was associated with renal dysfunction and the tolvaptan group has unfavorable background. 

    • Table 1: the AFP median value in the “Conventional Group” was 232 ng/ml while in the “Tolvaptan Group” it was 159 ng/ml. HCC was present in 33 patients of   the “Conventional Group” while it was present in 89 patients of the “Tolvaptan Group”.  How many patients had a previous diagnosis of HCC and how many patients developed HCC during the first six months of the follow-up period in the three different groups (i.e. Conventional Treatment, High-dose furosemide and Low-dose of furosemide treatment)? 

      →Thank you for suggestion. These are essential information. We added the number of patients with previous diagnosis of HCC and patients diagnosed HCC during the first 6 months of the follow-up period in the Table 1 (page 5).

    • At the time of enrolment, how many patients were in Child-Pug class B and how many in class C?

That’s true, and these information make it easy to understand the background for readers. We added the number of patients with Child-Pugh B and C in the Table 1 (page 5).

    • As reported by the Authors, the 1-year probability of survival after developing RA was to be approximately 30%. Would be, therefore, useful to highlight how many patients underwent to Transjugular intrahepatic portosystemic shunts and/or to liver transplantation during the follow-up period. Comments in this regard are meritorious and recommended in the revised version of the paper.

   These are good points, but in this cohort, there were no patients who underwent to TIPS or liver transplantation.

    • In the comparison of the cumulative survival rates between the “Conventional therapy Group” and 126 the "Tolvaptan Group" section,   Figure 3 showed  a survival period of 90 months.  The period of enrolment, however, was from 2010 to 2018. In  the reviewer opinion  would be better  to analyse and represent  the survival rates at 12 and 24 months for all patients. 

        →Thank you for suggestion. We added the sentences to show the survival rates at 12 and 24 months for all patients (page 5, lines 131-2).

Reviewer 2 Report

in this paper, Hosui and collaborators analyzed the effect of tolvaptan early administration to cirrhotic patients with refractory ascites, concluding that the use of this drug led to an increased survival in patients being treated with low-dose of furosemide. Thus, the conclusion of the authors is that the early administration of tolvaptan, rather than increasing the dose of furosemide or other loop diuretics, led to an increased survival. 

The study is interesting and scientifically sound, and an accurate statistical analysis has been performed. 

Overall comment: a comprehensive check of the English language should be performed.

Major comments:

The abstract should be reformulated in some parts (e.g. conclusion) to be more incisive.

Fig. 1: please organize and restyle better this figure for the sake of clarity. 

A short discussion about the cirrhotic patients with HCC may be interesting to add. Although this factor is not statistically relevant or different in the study groups, HCC patients could be characterized by a shorter lifespan compared to the other patients. Please comment on this.

Please stress the relevance of these findings for the clinical management of ascites in the conclusion, although considering the limited size of the study groups (this point should be mentioned ad a limitation of the study).

Please discuss about the possibility of designing dedicated not retrospective studies to confirm the clinical relevance of these findings. 

Minor comments:

some typos are present throughout the text. 

Author Response

Overall comment: a comprehensive check of the English language should be performed.

→That’s quite important. AJE (American Journal Experts) checked this manuscript again and it has been improved.  We got an editing certificate of this manuscript.

Major comments:

The abstract should be reformulated in some parts (e.g. conclusion) to be more incisive.

→Thank you for suggestion. We modified the abstract in order to be more incisive (page 1, lines 22-26).

Fig. 1: please organize and restyle better this figure for the sake of clarity. 

→Thank you. It is not easy to understand the scheme of this study. We changed Figure 1 in order to be more clear and easy to understand.

A short discussion about the cirrhotic patients with HCC may be interesting to add. Although this factor is not statistically relevant or different in the study groups, HCC patients could be characterized by a shorter lifespan compared to the other patients. Please comment on this.

→These are quite important points, and this manuscript might become interesting with the discussion about HCC. We added the sentences to discuss about patients with HCC (page 9, lines 212-220).  Overall survival among patients with HCC and without HCC in this cohort is investigated and found that it is significantly better in HCC (-) group than in the HCC (+) group (Supplementary figure 1). But the prognostic factor among patients with refractory ascites was gender (male), liver function (high value of total bilirubin), renal function (high value of BUN), and administration dose of furosemide (high dose of furosemide), but not the presence of HCC. Patients with refractory ascites have poor liver and renal function, and then prognosis might not depend on the presence/absence of HCC.

Please stress the relevance of these findings for the clinical management of ascites in the conclusion, although considering the limited size of the study groups (this point should be mentioned ad a limitation of the study).

→Thank you and these information is needed in this manuscript. We modified the sentence in the conclusion in order to stress the relevance of these findings for the clinical management of ascites (page 11, lines 256-262) .

Please discuss about the possibility of designing dedicated not retrospective studies to confirm the clinical relevance of these findings. 

→Yes, this is also important. Prospective and double blind studies should be better, and add the sentences about the possibilities of these studies (page 11, lines 260-2).

Minor comments:

some typos are present throughout the text. 

→We checked our manuscript again and corrected some mistyping (page 1, line 16).

Reviewer 3 Report

The authors present a retrospective cohort study and made subgroups of patients before and after adding tolvaptan to the treatment regime. There is no significant improvement of overall survival between groups. There may be a benefit of tolvaptan in the long-term in a low-dose furosemide subgroup.

General: 

  • the manuscript could be improved by following the reporting system of the STROBE guidelines for cohort studies. 

Introduction: 

  • no comments

Methods:

  • Not entirely clear what the primary outcome is and what the secondary outcomes are.
  • It seems as if there were no exclusion criteria, is this correct? Were there any patients that did nog receive tolvaptan after December 2013?
  • Due to its design, the study has a long duration. Were there any changes in protocols and standards during this time (besides addition of tolvaptan)
  • Was some kind of sample size calculation performed beforehand?
  • Why did one group have 3 years of inclusion and the other 5 years?
  •  Do you have data on occurrence of hepato-renal syndrome? 
  • How was multi-collinearity addressed (e.g. BUN and creatinin) ?
  • Has loss-to-follow-up been incorporated in the Kaplan-Meier curves? (censoring)

Results: 

  • were there differences in the use of paracentesis in the groups?
  • would suggest to not use hepatic edema in paragraph 3.3
  • Please also add an high-dose and low-dose comparison for the conventional group (like Table 4, and add lines to Figure 4). Comparing a (favorable) subselection from one group to the whole control group does not seem like a fair comparison. 

Discussion/Conclusion

  • it is concluded that survival was iproved, however, Figure 3 and its analyses fail to show this. The low-dose tolvaptan group should be compared to a low-dose conventional group for accurate comparison. 
  • the discussion could elaborate on other long-term studies, what were the results there and how are they different
  • Any advice for (prospective/randomized) follow-up studies?

Author Response

  • the manuscript could be improved by following the reporting system of the STROBE guidelines for cohort studies. 

→The STROBE guideline is quite important and we read the STROBE guideline again and modified some sentences in order to satisfy the guidelines, especially in the Patients and methods section.

Methods:

  • Not entirely clear what the primary outcome is and what the secondary outcomes are.

→That’s are one of the most important description. The primary outcome is overall survival and the secondary outcomes are the prognostic factors for survival and contributing factors to a good response to tolvaptan.  We added these sentences in the Patients and methods section (page 2, lines 63-4).

  • It seems as if there were no exclusion criteria, is this correct? Were there any patients that did nog receive tolvaptan after December 2013?

   →Yes, there were no exclusion criteria, but tolvaptan was treated after we got an informed consent.  All patients received tolvaptan after December 2013. 

  • Due to its design, the study has a long duration. Were there any changes in protocols and standards during this time (besides addition of tolvaptan)

   →There were no changes in protocols and standards during this observation period.

  • Was some kind of sample size calculation performed beforehand?

→We calculated sample size by Schoenfeld method, and substituted; observation period was 5 years, event rates were 0.5 (control) and 0.3 (tolvaptan), loss-to-follow-up was 0.1, the power was 0.8, and alpha was 0.05.  Sample size calculated in this method turned to be approximately 150, and the number of this real cohort was 206.

  • Why did one group have 3 years of inclusion and the other 5 years?

→Observation period of the control group was approximately 4 years, and that of the tolvaptan group was 5 years.  The oldest data which we can investigate was Jan. 2010, and as a result, follow-up period was a little different among these two groups.

  •  Do you have data on occurrence of hepato-renal syndrome? 

→Thank you for suggestion. The occurrence of HRS is quite important, and we add these data in the Result section (3.1 Changes in renal function after hospitalization for the first time in the conventional therapy group and tolvaptan group).  Definition of HRS is based on the review article in Journal of Hepatology (Angeli P, et al. 2019, 71, 811-22), and the occurrence of HRS was 58% and 11% during one year, in the control group and tolvaptan group, respectively (page 4, lines 117-8).   

  • How was multi-collinearity addressed (e.g. BUN and creatinine) ?

→We have to exclude the possibility of multi-collinearity in this study, and confirmed the result was correct in this manuscript.  We analyzed contributing factors to a good response (Table 2) and the prognostic factors for survival (Table 3) multivariately again. The value of creatinine seems to be correlated with the value of BUN, and then a multivariate analysis without creatinine was done, for example.  We found that these results were almost same as the result we have already shown.  PT activity, albumin and bilirubin also seems to be correlated.  We examined in the same way as described above, and found the result did not change.       

  • Has loss-to-follow-up been incorporated in the Kaplan-Meier curves? (censoring)

Results: 

    →Loss-to-follow-up has been incorporated in the Kaplan Meier curves. Four patients in the control group and seven patients in the tolvaptan group were lost to follow up, and the ratio was 5.3% and there were no differences in these two groups.

  • Were there differences in the use of paracentesis in the groups?

→Refractory ascites were removed by paracentesis in all patients when ascites were not under control and patients complained of feeling of fullness.  We have never used paracentesis if drugs are effective and ascites are under control.  There were no differences in the use of paracentesis between the control group and tolvaptan group (68% vs 59%, p=0.109).

  • would suggest to not use hepatic edema in paragraph 3.3

→We changed the word of hepatic edema to ascites (page 6, line 141).

  • Please also add an high-dose and low-dose comparison for the conventional group (like Table 4, and add lines to Figure 4). Comparing a (favorable) subselection from one group to the whole control group does not seem like a fair comparison. 

→That’s right and we split the conventional group into two groups (high-dose conventional group, low-dose conventional group.  Overall survival was investigated, but there were no differences between high-dose and low dose conventional group.  This result was shown in Supplementary Table 1 and Figure 2.

Discussion/Conclusion

  • it is concluded that survival was improved, however, Figure 3 and its analyses fail to show this. The low-dose tolvaptan group should be compared to a low-dose conventional group for accurate comparison. 

→The low-dose tolvaptan group was compared to a low-dose conventional group, and the result was shown in Supplementary Figure 3. It becomes more complexed to add 2 more lines (Figure 4) because Kaplan-Meier curves of low-dose and high dose conventional therapy group were almost same.

  • the discussion could elaborate on other long-term studies, what were the results there and how are they different

→Thank you for advice. These are also essential information. We added the sentences in the Discussion section (page 10, lines 245-53).  We have never found the study that the tolvaptan group and conventional therapy group had been compared in more than 200 patients with refractory ascites for more than 4 years.

  • Any advice for (prospective/randomized) follow-up studies?

   →Prospective and double blind randomized studies should be better, and we added the sentences about the possibilities of these studies in the conclusion section (page 11, lines 260-2). 

Round 2

Reviewer 1 Report

The Authors replied in a satisfactory way to all the comments raised by the reviewer. The paper is, therefore, recommended for publication.

Reviewer 3 Report

No further comments